# Branding Built Heritage through Cultural Urban Festivals: An Instagram Analysis Related to Sustainable Co-Creation, in Budapest

**Bálint Kádár * and János Klaniczay** 

Department of Urban Planning and Design, Budapest University of Technology and Economics,
1111 Budapest, Hungary; klaniczay.janos@urb.bme.hu
* Correspondence: kadarb@urb.bme.hu; Tel.: +36-20-410-28-58

**Abstract:** Global tourism is posing challenges on the environmental and social sustainability of host communities, while the industry itself has proven to be vulnerable to threats such as a global pandemic. Proximity tourism was demonstrated to be a more sustainable form in every aspect, especially when locals can co-create the experience and develop place attachment in urban environments through placemaking practices valuing previously underused urban heritage. An alternative urban festival in Budapest focusing on the built environment attracts locals annually to visit open houses providing visitors with genuine experiences. Residents are actively involved in the cultural placemaking practices of the event. As visitors documented the festival and the architectural heritage and uploaded hundreds of photos of their experience to social media platforms such as Instagram, they contributed to the branding process of the event and to the placemaking process involving less known heritage values. In this study, a dataset of more than ten thousand posts was retrieved by scraping Instagram posts based on hashtags related to the Budapest100 festival and analyzed from a temporal and spatial aspect. Returning visitors were identified, who contribute substantially to the sustainability of the event and to the branding of the built environment. Results suggest that community-based local urban festivals are a sustainable form of proximity tourism, resilient even to the COVID-19 pandemic. Place branding through urban festivals focusing on the local built heritage can also decrease the growing pressure on city centers in tourist-historic cities dealing with overtourism.

**Keywords:** proximity tourism; cultural urban festival; heritage branding; Instagram; Budapest

## 1. Introduction

Urban tourism is a growing phenomenon [1], but its sustainability has been questioned many times. During the COVID-19 pandemic that started in 2020, nearly all international tourism came to a halt, but as restrictions are gradually lifted around Europe, the subject of sustainability of urban tourism in tourist-historic cities is once again a trending topic. Meanwhile, there have always been sustainable processes of urban tourism such as including and involving locals in the creation of authentic tourist experiences. Proximity tourism as a new trend of locals participating in near-home tourist experiences has recently emerged [2,3] and has made the tourism industry more resilient towards the pandemic [4]. The branding process of the cultural heritage via social media [5] has also shown how grassroots methods of place branding can provide sustainable solutions.

When discussing the topic of sustainability in urban tourism, these small-scale, bottom-up local aspects should be emphasized, all of which can be found in a local urban cultural festival focusing on architectural and intangible heritage in Budapest, Hungary. During the festival, locals are involved as organizers of placemaking events creating the urban brand of previously unappreciated locations. Visitors are actively included in the co-creation

process of the experiences; furthermore, visible effects on international tourism can be noticed as well.

This study focuses on the branding process of the built environment via the social media interactions connected to the festival, its main goal is to find evidence on the community co-creation of such brands. The main research question is whether local visitors can effectively contribute to placemaking processes, expanding the domain of places of interest for proximity tourism by generating interest for previously undervalued built heritage sites. Most parts of this process are well-documented in theories and case studies, but very few are the studies delivering empirical evidence on the co-creation process based on the user data of those visitors who contribute in great numbers to the branding of built heritage. In the introduction the theoretical background of proximity tourism, placemaking, and local urban festivals from a sustainability aspect will be presented, additionally current trends seen between social media and urban tourism will be discussed as well. Section 2 will introduce the Budapest100 festival and describe the methodology of how data acquisition from Instagram was carried out. Results will be presented and discussed in detail in Sections 3 and 4, analyzing thousands of Instagram posts showing temporal and spatial trends of the festival. Theoretical and practical implications of the study will be included in Section 5, closing with concluding remarks.

### 1.1. Proximity Tourism, as a Sustainable Visitor Experience

The phenomenon of tourism is based on the experience to gaze upon out of the ordinary environments and encounter out of the ordinary activities [6]. However, the experience does not have to be geographically distant in order to be unique [3]. The increased awareness of the impact of tourism on the climate change and the recent COVID-19 crisis affecting tourism brought much attention to short distance tourism, called proximity tourism in an increasing number of papers [7]. Scholars, such as Jeuring and Diaz-Soria define proximity tourism as a form of tourism emphasizing short travel distances, local destinations, and therefore less carbon emissions by transport [8].

The COVID-19 crisis made the sustainability of proximity tourism more evident [9–11], as international travel halted almost completely. Local tourism practices were not only able to survive, but in some rural settings they could also thrive during this period [12,13].

However, proximity tourism was an important phenomenon already before the COVID-19 pandemic. An important aspect of such short-distance tourism is the fact that locals living in the area are the ones having the tourism experience. 'Tourists in their own city' are visitors gaining more and more attention in urban and tourism policies, branding and tourism research as well [2]. A distinction can be made between short term proximity tourism and longer-term vacations in proximity destinations, as well as between locals' preferences for tourism; many times, the social status and possibilities of locals define their willingness to participate in local tourism activities [14]. While proximity tourism still builds on the experience of the extraordinary versus the daily ordinary, the lack of cultural differences facilitate the local tourism consumption in communal forms, such as festivals, urban walks, or even Instameets [15].

Guided walking tours are a common practice in urban tourism, recently gaining more attention in academic research papers [16]. Studies highlight the role of guiding [17], as the guide acts as an interpreter for the built environment and can also enhance the authenticity of the tourist experience [18], which is something visitors require more frequently. In return, participants of walking tours often become more than simple spectators, and with their comments they co-produce the experience [19]. With the increasing popularity of walking tours, partly due to the emergence of free walking tours in all major tourist destinations, urban places become 'glocalised' [20], and tourists seeking authentic experiences search for tours in less frequented neighborhoods, where the specific knowledge and interpretation of the guide is essential. Even though these alternative urban walking tours are becoming more popular, for international tourists they only represent a niche market [21]. As Diaz-Soria [3] points out, local residents are quite interested in walking tours in their own

city, and tours with a principal target audience of locals are appearing all over large cities. Walking in urban environments has benefits for the psychological wellbeing as well [22], enforcing local identity and even a sense of community. This factor was very important during the COVID-19 pandemic when locals explored their own cities more often due to travel restrictions and the lack of tourists in historic centers. There is great potential in proximity tourism during and after the pandemic [4] as it is a resilient and sustainable form of tourism.

It must also be noted that proximity tourism makes the co-creation of tourist experiences between locals and tourists much easier, and the demand for more participative and interactive experiences is also an emerging trend [23]. Binkhorst and Dekker argue that co-creation in tourism results in a series of added values, and it is time to look at the human in the center of the tourism experience network, not separately to the local or tourist, redefining the host-guest relationships [24]. Another theorization comes from the notion of creative tourism [25], identifying creative interactions and different motivations to participate from the tourist's side [26]. In this paper, a well-documented case of co-creation in tourism will be analyzed, comprehending the involvement of locals into the organization of a festival, the interactive experiences through exploration and storytelling in heritage settings, and the co-creation of a tourism related brand on Instagram [27].

### 1.2. Urban Heritage and Placemaking Practices as a Resource for Sustainable Urban Tourism

Urban heritage is the main resource in cities to create a unique and sustainable local identity for local communities [28], to develop personal place attachment for community members channeling personal motivations into community goals [29], and to develop a unique and sustainable tourism offer that can enhance the economic and cultural possibilities of given community. Urban identity, place identity, and how these relate to the self-identity of a local [30,31] are topics treated by behavioral sciences since long ago [32,33]. Place attachment is important not only for the residents in order to value their neighborhood, but also for the visitors to build up loyalty for the destination and to have a meaningful tourism experience [34]. In fact, there is only a thin line between tourists and locals when the visitors of a local urban destination or a heritage building are discussed. According to Hoogendoorn and Hammett, 'resident tourists' within proximity tourism are the ones that can be involved the most in co-creation, as their 'self-branding' practices contribute to the branding of the heritage itself [15], and therefore to the building of a local identity [29].

Tourism, however, needs branding in order to differentiate the experience or destination from the ordinary and to indicate to the potential visitor what to visit. Following the semiotic of attraction theory of MacCannell [35], no sight can be a tourist attraction without markers giving a significance to it. Proximity tourism usually involves local destinations less branded for international tourism, but still the marking and branding process is needed in order to attract local visitors. Most sights in urban tourism are heritage items such as buildings, and as stated before, the integration of urban heritage into the self-identity and urban identity of locals is beneficial for local identity and the socio-economic life of cities. Thus, the marking and therefore branding of local urban heritage for proximity tourism is an important aspect in urban heritage management, but the tools of branding must be more inclusive and less resource-intensive than traditional tourism marketing tools. The abovementioned urban walks are one of these local inclusive tools, as the tour guide can effectively 'mark' the sights and involve the visitors on a personal level. The action to 'mark' the given urban heritage for tourism results in a meaningful place created from a previously ordinary and unimportant sight [36]. This process is called placemaking, and even though it has strong connections with tourism branding of urban heritage, it mainly aims to give back the significance and many times the use of an urban space to the community that can value its heritage the most [37]. In this paper, the effectiveness and sustainability of a community driven 'marking' process will be demonstrated.

### *1.3. Local Festivals as Sustainable Branding Practices for Tourism and Urban Heritage*

Several instruments are available for branding urban places, most commonly the visual qualities of the built environment are used in destination marketing, but cultural events are also an effective tool for branding [38], and as Ashworth [39] points out, combining these practices can be beneficial. Visitors attending a festival regularly develop a sense of place attachment and as a consequence the loyalty towards the festival can be transformed into loyalty towards the place [40], even if the theme of the festival is not related to the built environment [41]. During large scale urban festivals, the cityscape is transformed in a curated and controlled manner to serve the experience economy [42], while smaller-scale festivals in urban neighborhoods have a much more intangible community-making effect and cultural authenticity [43]. The added cultural value during a festival plays an important role in placemaking mechanisms as it can contribute to the local identity [44]. According to Lew [36] these cultural events can be considered as 'creative placemaking', but if we take into consideration the ephemerality of urban neighborhood festivals, we can argue that tactical placemaking aspects are equally relevant. In order to create a sustainable and repeatable event, the local community needs to be involved in the creation process [38], so that they are personally interested in the annual organization of the festival. The process is reciprocal, as place branding has community branding effects as well [45]. If the local cultural heritage is in the focus of the festival, it can promote local identity in the long run, which is also beneficial for city marketing strategies [46]. The described logic of place branding leads a growing number of experts to emphasize the importance and potentials of participatory cultural events and place attachment [47]. In this paper, such participatory festival and its effect on place branding is analyzed.

### *1.4. Social Media and Urban Tourism*

A main factor affecting tourism among other aspects of social life in the past decade was the spread of social media. Proximity tourism and co-creation processes were even more affected, as the visitors themselves can now contribute to the formation and image of a destination. Social media has become the main platform of destination branding, users sharing their experiences and influencing other users in their travel choices. Image and video sharing platforms gained the most important role.

Urry [6], Hall [48], and MacCannell [35] formulated the role of photography in tourism well before the digital revolution. The 'circle of representation' for tourist destination images described by Jenkins [49] building on the influential work of S. Hall [48] is more relevant than ever in the age of Instagram. According to this model, the tourist has an imagined picture of a destination already before its visit from images conveyed to him by marketing or other means (marketers). When arriving to the place, the tourist feels the need to take a photograph of the perceived image and share it with others, contributing to the image conveyed to other tourists. This process only got stronger with the digital and social media revolution, as imaging and sharing have become so widespread that there is a competition to publish new, unique, and yet attractive images. This has brought previously undiscovered sites into the public consciousness, but because of the functioning of the vicious circle described above, they have also become known and desired destinations [5,50]. In some cases, this circle of representation is recognized by Destination Management Organizations (DMOs) who use social media platforms like Instagram as a co-creation tool in tourism [27]. Today, significant research focuses on how social media affects the perception and development of tourist destinations [51–54]. Such research confirms that UGC (User Generated Content) on social media fundamentally influences the perception of a destination, and has a great marketing value especially for smaller, specialized businesses, mainly serving short visits [55]. Thus, these media and the short messages and visual content circulating on them promote the fragmentation of large tourism infrastructures and the viability of smaller-scale services and local attractions, favoring co-creation in tourism and reinforcing proximity tourism. One of the main fields where the use of social media is becoming a consciously used tool is heritage conservation [56]. Content co-creation together with visitors is an

emerging tool used by DMOs, heritage management organizations, or NGOs involved in heritage management.

Tourism consumption can only be fulfilled if evidence of the experience is produced and shared with others [6]. Social media helps this process [57], but also brings the tourism experience to a more mundane level. In the twentieth century 'Kodak Moments' were the highlights of once-in-a-lifetime vacation trips [58], while today we can find an "Instagram moment" anywhere [5], therefore long-distance travel is no longer necessary to post about any leisure activity that can become popular on social media. This favors proximity tourism, and the 'circle of representation' helps to promote nearby experiences to others, creating interest in visiting less marketable destinations, such as the less famous examples of the architectural heritage of an urban destination.

Platforms such as Instagram are not only tools to generate local tourist interest in less marketed architectural heritage [59] but are also great researchable databases to understand such mechanisms by scholars. There is currently no other accessible method for mapping the statistically invisible tourism of proximity tourism. Some social media services allow the geographical positioning of the uploaded photography. Location based tourism research is an emerging field in the past decades [60,61], and the 'geotags' of images uploaded to sites like Flickr.com (accessed on 17 March 2022) allow for such research on the spatial behavior of tourists [62–66]. Some studies focus especially on historic urban landscapes [67], but it must be noted that the new wave of social media platforms, such as Facebook or Instagram does not allow the direct acquiring of the geographical data of the uploaded photos, therefore the era of tourism geography research through 'geotags' is likely over.

However, the less transparent era of Instagram also has new possibilities because of the different content, and new research techniques emerged to harvest that content. Since its founding in 2010, Instagram became a daily image sharing tool for large portions of smartphone users, making these users interested in aesthetically appealing topics, such as architectural heritage [59]. User interaction then generates Likes, which is a good marker of the popularity of the photographed content [68]. While the first generation of social media had often an open Application Programming Interface (API), retrieving metadata from Instagram is only possible with solutions involving scraping of posts, based on a hashtags or other parameters [69,70]. Such a scraping technique is used for the database of this study, described in the next section.

## 2. Materials and Methods

Budapest is a tourist-historic city well known because of its views from the Danube River, but it also has the largest historical residential urban fabric from the end of the 19th and beginning of the 20th centuries, hard to value as a heritage asset because of the neglect in state socialism and its residential character [71]. Even though branding as a post-socialist city has not been flawless since the change of the regime in 1989 [72], the city has the resources to positions itself as a cultural capital [73].

Budapest100 is a civic festival in the Hungarian capital city launched in 2011 to celebrate the birthday of 100-year-old buildings attracting thousands of visitors with various programs [74]. Around 50 condominium buildings and institutions open their doors on this yearly weekend with hundreds of volunteers and the help of local residents. Visitors of the festival can enter open houses free of charge and are welcomed by cultural programs and exhibitions about the tangible and intangible history of the buildings. The typical inner courtyards of tenement houses built in the late 19th and early 20th centuries [71] provide ideal spaces for the festival, which can be conceived as a network of small-scale placemaking events [36]. On the one hand, the festival aims to show the diverse architectural heritage of the city to its own citizens; the official motto 'every house is interesting' points out how the intangible heritage tied to buildings can increase the value of even more mundane architectural heritage. On the other hand, the organisers institutionalized by the Hungarian Contemporary Architecture Centre foundation (KÉK) worked out professional methodologies to involve residents in the program, educating them about the values of the

built heritage they are surrounded with. KÉK foundation aims to create programs reinforcing the social sustainability of architectural heritage and the urban environment [75], and in a small scale the Budapest100 festival has a measurable impact in this, creating a sense of place to which residents can strongly relate their own self-identity [30,31]. In several cases new communities of residents were formed in the aftermath of the festival, as inhabitants owning the apartments in these condominiums did realise that their buildings are of interest to the hundreds of visitors and are worth taking care of. Thus, Budapest100 festival became a proven tool for placemaking [37,38], but the main research quest of this study is to empirically demonstrate the co-creation process and sustainability of the brand upscaling ordinary built heritage.

To answer the research question, Instagram posts related to this local cultural urban festival have been analyzed to quantitatively measure its local identity forming and brand making capabilities. Instagram was chosen for various reasons over other popular social media platforms such as Facebook which is not primarily for sharing photos, or Twitter which is not a prevalent platform in Central-Eastern Europe. Visitors and participants of Budapest100 take hundreds of photos each year during the festival and upload a selection of them to Instagram. The official account for the festival was created in 2016, after which its Instagram presence became more active, but even before this, photos taken during the festival were already tagged #budapest100. In certain years, thematic hashtags were introduced by the communication team of the festival to create a sub-brand for the year (Table 1), such as #bp100bauhaus and #budapest100bauhaus in 2019, when the theme of the festival was interwar modernist heritage of the city, connected to the Bauhaus centenary. Still, #budapest100 is by far the most popular hashtag used in relation to the festival, a total of 14,498 posts had this tag as of January 2022.

**Table 1.** Availability of hashtags to analyse in relation to Budapest100 festival and two cultural festivals for control analysis (as of January 2022). Source: Authors.

| Hashtag | Total Number of Posts as Indicated by Instagram | Public Posts Available for Scraping |
|---|---|---|
| #budapest100 | 14,498 | 10,527 |
| #budapest100rakpart | 177 | 129 |
| #bp100bauhaus | 401 | 304 |
| #budapest100bauhaus | 141 | 99 |
| #bp100ujratervezes | 63 | 63 |
| #budapest100ujratervezes | 183 | 179 |
| #budapestitavaszifesztival | 504 | 361 |
| #belfeszt | 2976 | 1571 |

Unlike photo sharing sites like Panoramio or Flickr, which have an open API [61,66,67], Instagram offers no interface to make user data accessible for research. Some researchers rely on third-party Web-crawling services such as Magimetrics [70] or 4K Stogram [68], who have API agreements with Instagram to access their database, but these services have high fees. The other popular approach is to scrape the social media platform. The dataset of this study was scraped from Instagram on 7 January 2022 using a python script based on an open-source code shared on GitHub [76], originally created for the research of city parks in Bratislava [69], a study that inspired this paper. A new account for the purposes of this research was created, so no prior influence from the algorithm of Instagram would corrupt the search. The scraper script was used to extract all public posts tagged with #budapest100. Other related hashtags of the festival and some control data of similar festivals from the city were also downloaded. The download for each hashtag resulted in multiple json files, where each single post was affiliated with nearly 200 lines of information. A postprocessing script created a single json file containing only the following data fields for each post of the downloaded hashtags:

- Post ID
- URL of post
- Username

- Full username (if provided)
- User ID
- Timestamp
- Number of likes
- Number of comments (if any)
- Number of photos in carousel
- Location name (if geo-tagged by user)
- Latitude and longitude coordinates (if geo-tagged by user)
- Text of the post
- Hashtags used in the text of the post

For the budapest100 hashtag, a dataset containing a total of 10,527 posts was downloaded and a further 2706 more posts were also downloaded from thematic and control hashtags. The 10,527 #budapest100 posts were created by 2966 users over the course of a decade since 2012, containing a total of 17,067 photos (Instagram posts may include multiple photos in a carousel). These posts generated 36,579 interactions in comments and 1,338,558 likes on the posts. To analyze this large dataset the analytics software Tableau was used.

## 3. Results

### 3.1. Temporal Analysis of Posts and Likes

A line graph based on the timestamp of each post was generated to reveal the temporal trends of the #budapest100 hashtag, as shown in Figure 1. Likes have been also counted, measuring the popularity of the posts and of the hashtag. Data for further 2706 posts was extracted using the same scraping technique for comparative purposes (see Table 1), visualizing the posts of two similar, but unrelated festivals, and the complementary hashtags of the Budapest100 festival as well. The various hashtags related to the thematic years of the Budapest100 festival were used either together or separately from the original hashtag. Visitors were asked to use these tags in their posts by the organizers: 'rakpart' in 2017, 'bauhaus' in 2019, and 'újratervezés' in 2021.

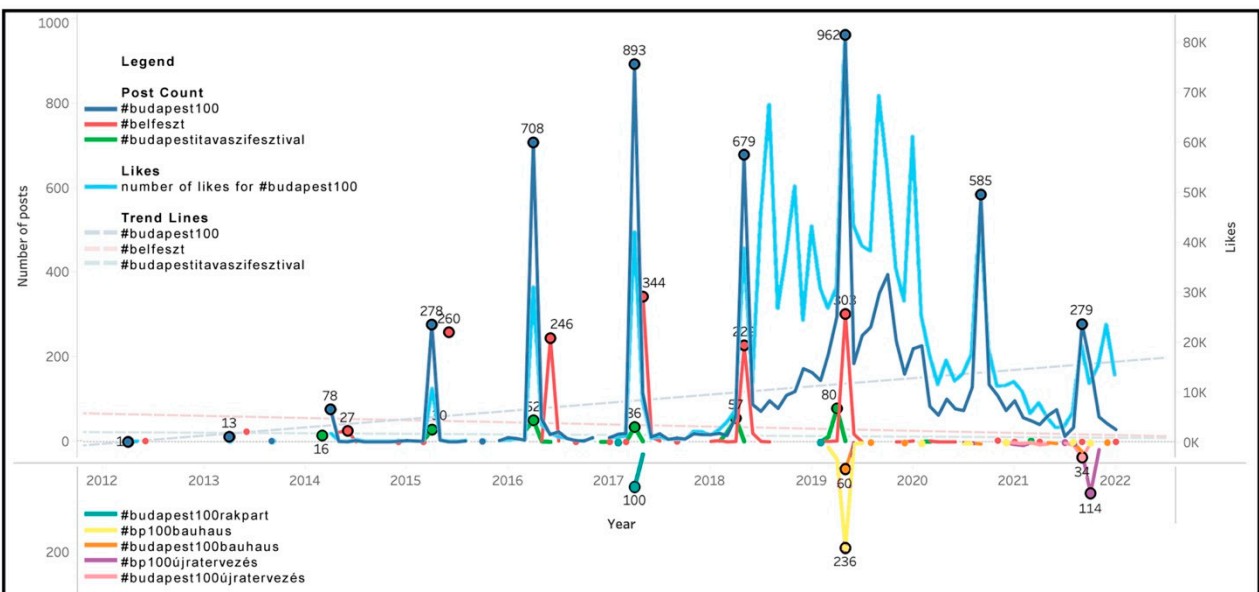

**Figure 1.** Number of posts and likes for #budapest100 hashtags, compared with the number of posts of two other urban festivals from Budapest on Instagram between 2012–2022 with trend lines. Related sub-brand hashtags of thematic years with peaks highlighted at the bottom. Source: Authors (retrieved from Tableau).

All of this data was aggregated and visualized for each month (Figure 1). The peaks found in the graph correlate with the dates of the festival. It is evident that likes and posts started to be well distributed all year round since 2018, as this was the year when organizers of Budapest100 started to use the Instagram platform as a year-round communication tool instead of just a photo sharing possibility. However, data presented in Figure 2 show that also other users and tourists started to post with #budapest100 hashtag outside the festival month starting in 2018. The festival was traditionally held in April or May, but in 2020 and 2021 the event had to be postponed to early Autumn due to the COVID-19 pandemic, as restrictions were only eased during the Summer and early Autumn months in both years. It is evident that the pandemic drew back the number of posts and likes of the festival; a significant backdrop is visible in early Spring 2020, but Budapest100 remained active as opposed to the two well established festivals from Budapest, chosen as control measures.

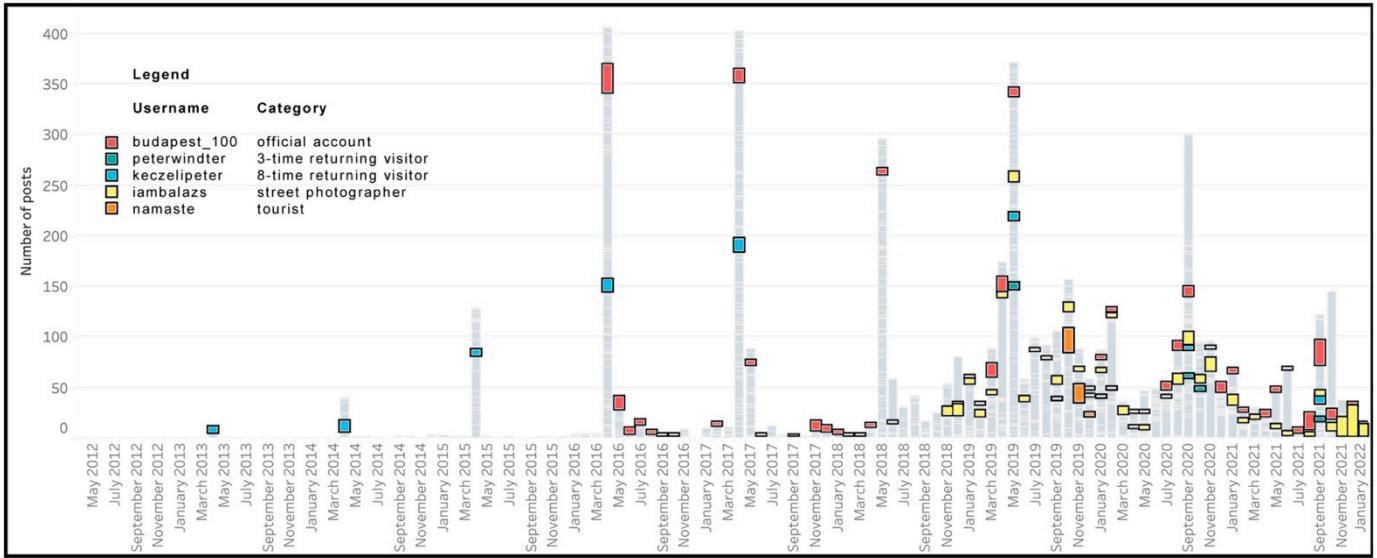

**Figure 2.** Monthly count of posts for every user posting 10 or more photos over the period of 2012–2022 using #budapest100 on Instagram. Highlighted patterns: The official Budapest100 account; three-time returning visitor; eight-time returning visitor; unrelated frequent Instagram user; occasional tourist. Source: Authors (retrieved from Tableau).

The posts under #budapestitavaszifesztival gather photos from the Budapest Spring Festival (BSF), a cultural festival organized by the municipality of Budapest since 1981. The #belfeszt tag represents posts from Inner City Festival (ICF), a free urban music festival organized by the municipality of District V (Inner City) annually since 2007. Both events show similarities to Budapest100 as annually reoccurring cultural urban events. The ICF as an annual free event, taking place in the center of the city draws massive crowds. Even though the focus of the festival is not the built heritage, festivals in urban settings have a similar effect of place-making [41], and hundreds of posts are generated every year. The brand of BSF was already well-established by the time Instagram became one of the principal virtual spaces to create brands, so not a lot of effort was put into the use of social media for the communication of the festival. In both cases, the Spring event had to be postponed since 2020 due to the COVID-19 pandemic, and almost no social media activity can be seen. A linear trend model was computed in Tableau for the number of posts in each month, comparing the trends of the three festivals (Figure 1). The regression model was statistically significant ($p = 0.0097$), since more than 10,000 values were iterated. From the trend line, it became evident that even with the number of posts decreasing in the past years the Budapest100 brand has a positive tendency, which cannot be told of the other two festivals.

### 3.2. Finding Returning Visitors through Their Posts

The Budapest100 brand on Instagram was built not only by the consistent use of #budapest100, but also by users returning as a visitor to the festival each year. Organizers of the festival were always aware of the fact that the audience of Budapest100 have many returning visitors. The volunteers in the organizing team often come from visitors of previous years. After analyzing the downloaded data certain patterns of returning visitor behavior could be observed.

In order to identify returning visitors, users who posted on Instagram with #budapest100 between 2012 and 2022 were categorized based on the distinct count of years when they have created content. Since only one festival was organized each year, this measure is fitting to the question. A total of 2538 users (85% of all users) only created posts in one year, and therefore were not identified as returning visitors. A total of 282 users were two-time visitors, 80 three-time visitors, 32 four-time, 21 five-time, 9 six-time, 1 seven-time, and 3 eight-time returning visitors were identified (Table 2).

**Table 2.** Returning visitors of Budapest100 based on the use of #budapest100 on Instagram. Source: Authors.

| Number of Returns | Number of Users |
|---|---|
| 2 years | 282 |
| 3 years | 80 |
| 4 years | 32 |
| 5 years | 21 |
| 6 years | 9 |
| 7 years | 1 |
| 8 years | 3 |
| Total number of returning visitors | 428 |

Another measure of returning visitors was the filtering of data for users with 10 or more posts. Browsing in the data we can notice certain patterns of user behavior highlighted in Figure 2. Regular returning visitors can be identified, who post only during the festival month (highlighted in green and light blue). The posting frequency of the Budapest100 official account can also be traced, showing how the communication has a periodicity, posting more frequently in the months leading to the event (highlighted in red). Certain users can be identified who use the hashtag monthly for their posts, unrelated to the festival, but still about the built heritage of the city (highlighted in yellow). Some users only use the hashtag during a short period of time, in between festival dates. It can be safely assumed that these users are tourists (highlighted in orange).

### 3.3. Spatial Analysis of Posts

The analysis of geo-tagged photos in tourism research is becoming a common practice [60] and Instagram posts may also include geographic information. It is an optional but easy step to add a geo-tag to an Instagram post on a smartphone, but this datatype is not as precise as automatic geo-tagging, as no precise location can be added, only a pre-defined site from the location database (automatically showing the possible selections as a list during the creation of the post). This location database of the 'Meta' company used also in Facebook contains streets, venues, or larger units like cities, each having coordinates, therefore allowing the analysis of the posts from a spatio-temporal aspect. Out of the 10,527 analyzed posts 73% (7642 posts) included geo-spatial information.

Since Budapest100 locations are usually residential buildings, the tags often refer to street names or districts close to the open house. A total of 35% of tagged posts were located at "Budapest, Hungary" (2366 posts) or simply "Budapest" (335 posts); these were excluded from the analysis. The map in Figure 3a shows the distribution of posts taken during the festival days, including two days before and five days after the event. This figure is showing clusters of posts being further from the touristic center of the city. A map of the photos taken by tourists at tourist locations was also created (Figure 3b), by filtering

posts from June 2018 until March 2020, excluding May 2019 (the month of the festival in that year) and posts by the official Budapest100 account. When comparing the two maps it is clearly visible how the tourist spaces are concentrated in the more central parts of the city and the Városliget (City Park), and how the festival changes the perception of heritage spaces in the city.

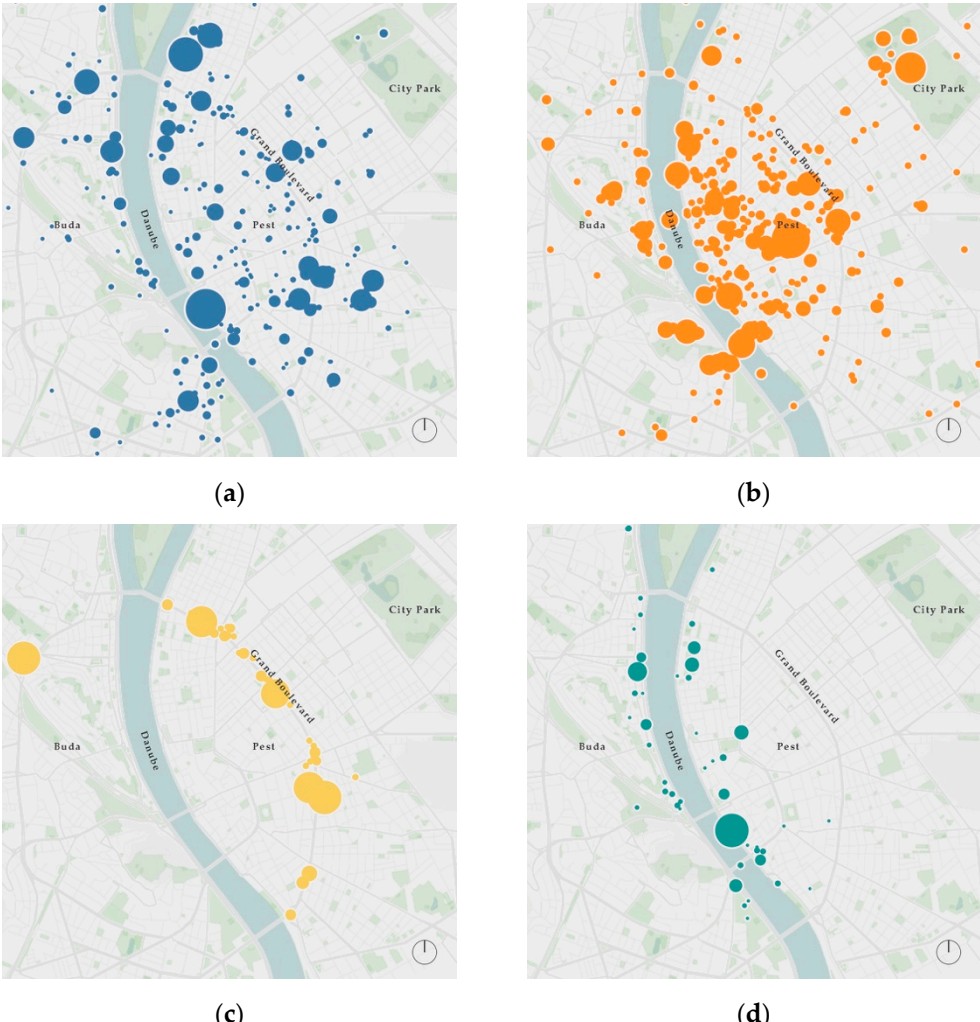

**Figure 3.** Spatial distribution of posts using #budapest100: (**a**) Posts created during the festival days with 2 days before and 5 days after included. Popular tourist locations excluded. (**b**) All posts created between June 2018 and March 2020, excluding month of festival in May 2019, and excluding Budapest100 account. (**c**) Posts created in 2016 during festival days with 2 days before and 5 days after included. (**d**) Posts created in 2017 during festival days with 2 days before and 5 days after included. Source: Authors (retrieved from Tableau).

The festival focused on large urban entities, the Grand Boulevard of Pest in 2016 and the embankments of the city in 2017. These special years meant that the heritage spaces of the city were thematized to certain locations during the weekend. This is quite visible in the maps showing the posts created during the 2016 and 2017 festivals in Figure 3c,d.

## 4. Discussion

Data of 10,527 posts containing 17,067 photos tagged with #budapest100 created by 2966 users on Instagram between 2012 and 2022 have been downloaded and analyzed. The development of the Budapest100 brand can be traced through the visualization showing the temporal distribution of the number of posts and likes connected to the festival (Figure 1). The first Budapest100 festival was held in 2011, while the first Instagram post from the

festival was created one year later. Since then, Budapest100 became a brand and the Instagram hashtag related to it became an effective tool to build that brand. Since 2015, the posts and likes related to Budapest100 were comparable to the largest urban festivals in the Hungarian capital, while the number of posts more than doubled in the following year, from 278 in 2015 to 708 in 2016.

The year 2016 was the first edition of the festival not concentrating on 100-year-old houses, but rather on an urban entity, the Grand Boulevard of Pest, while keeping the established brand name. The official Instagram account of Budapest100 was created in 2016 with the intention to help the branding of the festival on the social media platform [5]. The use of #budapest100 was therefore promoted by the organizers since 2016 and was no longer an organic phenomenon. The following edition focusing on the embankments by the Danube River became even more popular, with a total of 893 posts in April 2017, also having a sub-hashtag, #budapest100rakpart for the first time. These accompanying thematic hashtags of Budapest100 festival enforced the validity of the brand itself. The tag with 'Bauhaus' in it, referring to the modernist heritage in focus of the 2019 festival was also used, in some examples even years after the festival, showing the strength of the brand. The most popular festival according to Instagram was indeed in 2019, when the festival was going 'in the footsteps of Bauhaus' focusing on the interwar modern heritage of the city with 962 posts using the #budapest100 hashtag. Similar results can be deducted when inspecting the count of likes accumulated under each post tagged with #budapest100 in Figure 1. The most popular month is again May 2019 totaling at 80,420 likes.

Figure 1 shows how since 2018 the hashtag started to be used in the time between the annual festival weekend as well, due partly to the fact that the Budapest100 project started to communicate all year, but also in large part thanks to tourists using it. Visitors to Budapest not participating in the festival's program started using the tag to generate more traffic on their posts. After eight years of organizing an annual urban festival, the brand started to become more famous among Instagram hashtags than the festival itself. The consistent use of the well-established brand name also paid off, which is evident if we compare the number of followers of the official Instagram account and Facebook page of Budapest100 and that of the organizer, KÉK (Table 3). The festival has more followers than the organizing institution and stands as a brand on its own.

**Table 3.** Number of followers of Budapest100 and the organizer (KÉK) on various social media platforms (as of January 2022).

| Name | Instagram Follower | Facebook Page Followers |
|---|---|---|
| Budapest100 | 6007 | 27,277 |
| KÉK | 2325 | 14,160 |

As the festival's tag started to be used more frequently after 2018 the content of the posts started to recede from the open houses theme of the festival, as most users posting during the year were tourists. The budapest100 hashtag was used among other trending tags and started to appear under photos depicting tourist sites, or posts of travel agencies promoting travel to Budapest. However, the photos were still connected to the built heritage. In 2019 a secondary peak in September and then a third at the end of the year can be observed. These are months with strong city tourism in Budapest, while Budapest100 had no official activity. Such results show how visitors interested in the less known built heritage of a city effectively co-created a brand promoting this heritage [23,24,27]. The brand was later used by other visitors and tour operators valorizing this previously unexplored heritage of the city.

The sudden drop in early 2020 correlates with restrictions being imposed on travel due to the COVID-19 pandemic. The 10th and 11th editions of the festival had to be postponed to early fall due to the lockdowns imposed from March to June 2020, and from November 2020 to May 2021. The inconveniences of the COVID-19 pandemic are visible in the lower numbers of Instagram posts; however, the organization of the festival was

successful, building on the opportunities that proximity tourism offered even when larger scale tourism was completely on halt [4], demonstrating the resilience of proximity tourism, but also the resilience of the brand.

Identifying returning visitors (Figure 2) and analyzing certain patterns of user activities on Instagram brought important findings. Returning visitors contributed substantially to the sustainability of the festival, even during the pandemic. While the 2019 festival focusing on modernism was the most popular year based on the number of posts and likes (Figure 1), 2016 and 2017 were more popular among returning participants, and also in the COVID-19 affected editions, such visitors remained loyal, showing how place attachment formed during the previous festivals and grew stronger as place loyalty [40], and in this case brand loyalty as well. The popular 2019 festival in fact attracted more one-time visitors and with its focus on modernist heritage instead of historical architecture it was able to reach out to new audiences, showing the opportunities of community festivals and social media in heritage conservation [56].

The spatial distribution of posts revealed how the festival was able to extend the spatial system of tourism in Budapest [62], and thematize the urban space at the same time. Thematic years during which grand urban ensembles were in the focus of the festival (the Grand Boulevard in 2016 and the Danube embankment in 2017) the post locations show clear correlations with these urban ensembles (Figure 3c,d). Visitors of the festival have 'put on the map' the urban heritage that had opened up for them, and when the theme of the festival was not tied to a spatial system, the posts revealed a uniform distribution in space all over the historic urban landscape (Figure 3a). In contrasts, the posts created mainly by tourists in the remaining months of the year (Figure 3b) showed a much more concentrated distribution inside the most well-known tourist districts of the city, branding also popular tourist hotspots like the City Park, where the Budapest100 festival never had activities. Such finding did provide relevant insight into how a local urban festival involving placemaking methodologies can change tourist space usage of historic cities [66], offering an alternative to overtourism by distributing visitors interested in cultural tourism in larger areas.

## 5. Conclusions

This study aimed to get a better understanding of the dynamics of sustainable place-branding via cultural urban festivals and social media. Budapest100, a local urban festival in Hungary focusing on the built heritage was chosen as a case study, as this annual event has successfully attracted thousands of visitors during the period of more than a decade, therefore a significant amount of data was available for analysis. Furthermore, the authors were familiar with the festival and the city, so first-hand experience was given on the subject.

The paper brought evidence for a series of theories and processes introduced in the first chapter. Budapest100 festival focusing on the built heritage attracted local visitors who behaved like tourists in a near-home environment [2,3]. Proximity tourism remained resilient against the COVID-19 pandemic. It is also important to highlight how this festival offered many opportunities for interactions between visitors and organizers [18], in fact often the residents themselves were volunteering to organize the festival in their opened houses. During this community-based event the co-creation of the visitor experience [19] lead to a more sustainable process as the place attachment of both the residents and the visitors grew [31], creating the opportunity for the birth of a sustainable community who is interested in and committed to their built heritage and will take better care of it.

The festival used creative placemaking practices [36]—small-scale events, concerts, workshops, building visits—to attract the attention of the visitors to the built environment. This case study showed how architectural heritage of tourist-historic cities is a resource for urban tourism that can be presented and branded in a sustainable form by placemaking methods. The framework of a local cultural festival can provide the ideal setting for such small events to take place as part of a system, creating a sustainable way to brand the

urban heritage. The branding itself is done by the visitors when they take photos of their experience [6] and post it on free photo-sharing social media sites, such as Instagram [5,27]. This study brought evidence from Instagram users on the co-creation of the branding process of the built environment during a local urban festival. This result led to a better understanding of the co-creation process itself, explained on Figure 4. Due to the self-generating nature of social media 'buzz', this process should be considered a sustainable tool for branding the built heritage.

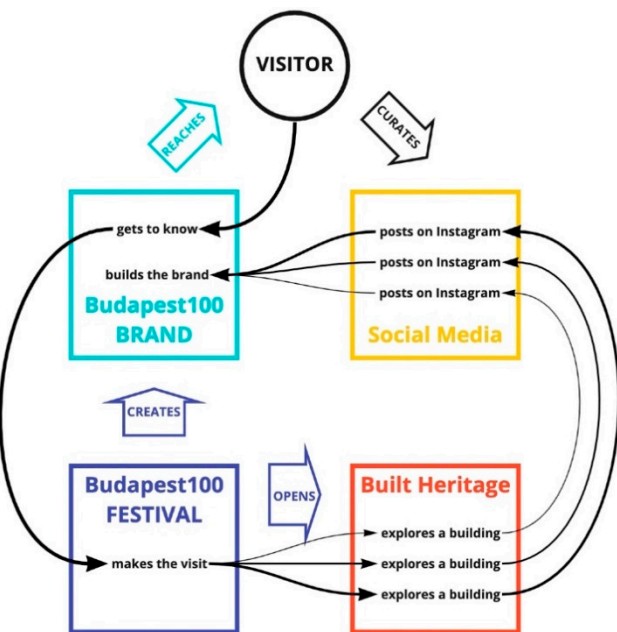

**Figure 4.** Diagram of the dynamics of place-branding via a local urban cultural festival and social media. Source: Authors (retrieved from Miro).

Due to the user base of this social media, the study has its limitations. Instagram users do not represent all age-groups, younger generations are much more involved, while older age groups are interested in urban heritage and in the festival at least as much as young people. Among them it is more difficult to research the consciousness of the use of the Budapest100 brand. The effect of social media on the visitor numbers of the festival could be measured, but the effects of a brand built on social media are hard to quantify on the ordinary uses of the built heritage, or on the normal tourism visits to the branded sites—more comparative data from other data sources should be used to verify the extension of the tourist space system at long term. This study could neither investigate the formation of place identity among visitors and organizers of the festival; therefore, questionnaires and further qualitative research methods should be used in the future to reveal deeper correspondences. Furthermore, the inspected local urban festival is a curiosity in regional scale, so even though other similar festivals from the same city were examined, other comparisons are necessary in regional scale to broaden the validity of the findings. Further research should be conducted with the same methodology described in this paper in other cities to provide valuable comparisons.

This paper is a valuable addition to the academic research of community-based urban festivals and how branding the built heritage can provide sustainable opportunities for regular and proximity tourism as well, even during a pandemic. More sustainable urban destinations can be developed with the community-based valorization of urban heritage, and such processes have no limits of growth according to the motto of Budapest100, as: 'every house is interesting'.

**Author Contributions:** Conceptualization, validation, writing, and supervision B.K.; methodology, formal analysis, data curation, visualization, and writing J.K. All authors have read and agreed to the published version of the manuscript.

**Funding:** The APC was funded by Budapest University of Technology and Economics, Faculty of Architecture 'Jövő Alap' Fund.

**Institutional Review Board Statement:** Not applicable.

**Informed Consent Statement:** Not applicable.

**Data Availability Statement:** Not applicable.

**Acknowledgments:** The authors would like to thank S. Márton for the technical support for running the python script.

**Conflicts of Interest:** The authors declare no conflict of interest.

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
