# Peer review of "Branding Built Heritage through Cultural Urban Festivals: An Instagram Analysis Related to Sustainable Co-Creation, in Budapest"

_sustainability, doi:10.3390/su14095020_

Round 1

Reviewer 1 Report

Congratulations to the authors! It's an interesting and relevant academic work.

I suggest that you do a full review of the article, as far as English is concerned. Some sentences are not correctly formulated, perhaps due to the direct translation that was made from another language to English.

And, after reading all the article I suggest that the authors rename it. Instead of "Branding architectural heritage by cultural festivals: an Instagram analysis of sustainable urban co-creation", perhaps could be something like that: Branding built heritage through cultural urban festivals: an Instagram analysis related to sustainable co-creation, in Budapest.

Some suggestions of improvement:

1. Introduction

  • Line 31 - During the COVID-19 pandemic started in 2020 ...
  • What is the main objective of this study? The authors do not describe it clearly, in Introduction part.

2. Literature review

  • The Literature review should not be in the Introduction. It must be a different chapter.
  • Line 57: you have 1.1 Proximity tourism, as a sustainable visitor experience. I think you should have 2. Literature review, then 2.1 Proximity tourism as a sustainable visitor experience
  • Line 58/59: the sentence is not correctly written (see the translation in all document).
  • Line 121 - The authors use de concept of "resident-tourists". Isn't this concept a little antagonistic? I think you should explain this a bit better, or add some references, because tourist definition implies a stay outside your residence for more than 24 hours. Maybe visitor concept is more suitable. 

3. Materials and methods

  • In this part of the article, the authors should explain better why they chose to analyze posts from Instagram and not from another social media network, like Facebook or Twitter.
  • Were the authors inspired by any other study that used a similar methodology?

4. Results

  • The authors start the topic with Figure 1. They should first put some text, with some explanation and then put the image.
  • In source of Figure 1 (and other figures), the authors should indicate the software from which they took the image. Source: Authors (retrieved from ....)
  • Line 328 - if you use Autumn (with capital letters), you should also use capital letters in lines 329 (summer and autumn) and 331/344 (spring).

Author Response

Thank You very much for this detailed review!

We went through all points, and tried to implement most suggestions, making also a proof reading for improved English:

Title: we changed the title to 'Branding built heritage through cultural urban festivals: an Instagram analysis related to sustainable co-creation, in Budapest.'

  1. Introduction
  • Line 31 - During the COVID-19 pandemic started in 2020 - corrected
  • What is the main objective of this study? The authors do not describe it clearly, in Introduction part.  We made profound revisions here, and complemented the introduction, all of its subchapters and also the conclusions. we hope it is clear now.
  1. Literature review
  • The Literature review should not be in the Introduction. It must be a different chapter. Line 57: you have 1.1 Proximity tourism, as a sustainable visitor experience. I think you should have 2. Literature review, then 2.1 Proximity tourism as a sustainable visitor experience  -> We did not change this, even if the suggestion was very logical. Most of the time papers published in Sustainability have a literature review in the introduction part, rarely there are separate chapters. In our case we introduce the sub-topics, not only make a literature review, so we followed this extended Introduction structure.
  • Line 58/59: the sentence is not correctly written (see the translation in all document). - We corrected this
  • Line 121 - The authors use de concept of "resident-tourists". Isn't this concept a little antagonistic? I think you should explain this a bit better, or add some references, because tourist definition implies a stay outside your residence for more than 24 hours. Maybe visitor concept is more suitable.  -> We made a more reference to the original author we cited, this is a concept explained in the reference, therefore we cannot change its name, which is already in 'parenthesis'
  1. Materials and methods
  • In this part of the article, the authors should explain better why they chose to analyze posts from Instagram and not from another social media network, like Facebook or Twitter. -> We did so
  • Were the authors inspired by any other study that used a similar methodology? -> we made more clear our references
  1. Results
  • The authors start the topic with Figure 1. They should first put some text, with some explanation and then put the image. -> We did this change
  • In source of Figure 1 (and other figures), the authors should indicate the software from which they took the image. Source: Authors (retrieved from ....) -> We did so
  • Line 328 - if you use Autumn (with capital letters), you should also use capital letters in lines 329 (summer and autumn) and 331/344 (spring). -> we corrected this

Thank You again

Reviewer 2 Report

Peer review

Title: Branding architectural heritage by cultural festivals: an Instagram analysis of sustainable urban cocreation  

Authors

Bálint Kádár and János Klaniczay

Introduction

Please update the literature review to 2021

The scope of the article is not very clear expressed

Figure 3. Please insert the legends to the maps. The maps also need to have some toponimes and orientation elements. Please revise

References

Please cite also:

 IlieÈ™, D.C.; Hodor, N.; Indrie, L.; Dejeu, P.; IlieÈ™, A.; Albu, A.; Caciora, T.; IlieÈ™, M.; Barbu-Tudoran, L.; Grama, V. Investigations of the Surface of Heritage Objects and Green Bioremediation: Case Study of Artefacts from MaramureÅŸ, Romania. Appl. Sci. 202111, 6643. https://doi.org/10.3390/app11146643

Great success!

Author Response

Thank You for your review.

We answer your comments here:

Please update the literature review to 2021 -> We included many literature from 2020, 2021 as well, we feel the literature in this topic relevant as it is, also because the concepts explained are better cited with well established papers, not necessarily the most recent ones. 

The scope of the article is not very clear expressed -> Thank You, we made a great effort to make a clear research question and good scope, inserting many sentences in the Introduction part, but also reflecting on these in the conclusion part!

Figure 3. Please insert the legends to the maps. The maps also need to have some toponimes and orientation elements. Please revise -> we did so, thank You

References, Please cite also: -> Thank You for the suggestion, we saw where this could fit, but the topic is different, we will see it for next time!

Reviewer 3 Report

This is an interesting study and in general the paper is well written. However, some key elements of the argument should be refined to strengthen the paper’s message and contributions. Right now, a lot of these remain implicit.

The paper and the study are interesting but the introduction should make a stronger case for:

  1. a) the gaps in knowledge driving this study;
  2. b) what the key objectives and research questions of the study were; and
  3. c) how this work contributes to knowledge, especially in relation to point a.

Similarly, the literature review provides a whirlwind overview of topics underpinning the study. However, there is no sense of the gaps in the knowledge or the problem that justifies and drives the current study. This should be outlined throughout the review, but the end of this section should also have a stronger synthesis that makes a case for the gap in knowledge and what this study will contribute to this body of knowledge.  

On page 6, a research question suddenly appears. However, the question is not phrased clearly. It is asked as a closed question. It is also not something that can be answered meaningfully or definitively on the basis of the study design. Plus, the underpinning justification for this question is unclear. I recommend revising this part, articulating it as several, open ended questions that may be a better it for this type of exploratory study. The research question should be linked to and justified by the literature.

The Discussion was somewhat disconnected from the literature review. The findings were interesting but they could have been interpreted further in dialogue with the literature. Having a stronger sense of the research problem at the start of the paper, and in the literature review, would help the discussion speak more clearly to the problem, which would help to articulate the paper’s contribution too. This was only really attempted in the final few sentences of the Discussion.

Having a stronger sense of the problem and questions from the start, and as a thread throughout, would also help to develop the conclusion so it speaks more explicitly to how this study has contributed to knowledge.

Author Response

Thank You for Your detailed review. 

We tried to answer most of Your concerns:

The paper and the study are interesting but the introduction should make a stronger case for: 

  1. a) the gaps in knowledge driving this study; 
  1. b) what the key objectives and research questions of the study were; and 
  1. c) how this work contributes to knowledge, especially in relation to point a. 

  -> We did follow Your recommendations, and reformulated the key objectives and the research question, making it clear in its place. New lines were inserted, not only in the Introduction, but in Methods and in conclusions, especially for point c)

Similarly, the literature review provides a whirlwind overview of topics underpinning the study. However, there is no sense of the gaps in the knowledge or the problem that justifies and drives the current study. This should be outlined throughout the review, but the end of this section should also have a stronger synthesis that makes a case for the gap in knowledge and what this study will contribute to this body of knowledge.   -> Thank You, we did so, added lines to explain where we will add to the theories reviewed, hopefully we made ourselves more clear now

On page 6, a research question suddenly appears. However, the question is not phrased clearly. It is asked as a closed question. It is also not something that can be answered meaningfully or definitively on the basis of the study design. Plus, the underpinning justification for this question is unclear. I recommend revising this part, articulating it as several, open ended questions that may be a better it for this type of exploratory study. The research question should be linked to and justified by the literature. 

-> we did just that. We changed that one research question, put a relevant part in the Introduction and reflected on that on the place of that question on page 6. we articulated our questions better we hope.

The Discussion was somewhat disconnected from the literature review. The findings were interesting but they could have been interpreted further in dialogue with the literature. Having a stronger sense of the research problem at the start of the paper, and in the literature review, would help the discussion speak more clearly to the problem, which would help to articulate the paper’s contribution too. This was only really attempted in the final few sentences of the Discussion. 

-> We added many reflections to the discussion part, thank You. I hope it became more consistent, in line with the topics in the introduction

Having a stronger sense of the problem and questions from the start, and as a thread throughout, would also help to develop the conclusion so it speaks more explicitly to how this study has contributed to knowledge. 

-> Thank You, we tried to do exactly that. Hopefully this version is more comprehensive thanks to Your observations and some greater review.

Reviewer 4 Report

Research methodology needs to be improved. It is too detailed.

There are no specific goals or hypotheses.

Please describe the test method. What is the international significance of the presented research?

Please provide the theoretical and practical implications of the research results.

What were the limitations in the implementation of the research, apart from the age of the respondents, about which the authors wrote. 

Author Response

Thank You for Your suggestions. We tried our best to answer these, and we believe the paper is much more comprehensive now.

There was one part we felt no need to change that much:

Research methodology needs to be improved. It is too detailed. -> We made some improvements, but did not feel the methods part is too detailed. Both the base data about the festival - subject of the case study - and the methodology on Instagram scraping is essential to make this study understandable and reproducible. The details on the data retrieved may seem too detailed, but so much detail is indeed needed in order to assure other researchers can follow and reproduce if needed the methodology.

There are no specific goals or hypotheses. -> Thank You, we agree You were right, we put an effort to be more clear on this. we put a clear research question, some scopes, more explanation on how it connects to the literature review, and more explanation on that for the Discussion and Conclusion chapters.

Please describe the test method. What is the international significance of the presented research? -> We tried to integrate the answer of this questions to the new parts I described in the answer to the previous question.

Please provide the theoretical and practical implications of the research results. 

-> we added some lines, and hopefully now the implications of the research results are more understandable!

What were the limitations in the implementation of the research, apart from the age of the respondents, about which the authors wrote. 

-> Thank You, we added more to that part,

Thank You again, we hope the paper got improved because of these changes!